

# Morphospace saturation in the stem-gnathostomes pteraspidiformes heterostracans: an early radiation of a 'bottom' heavy clade

Marco Romano[1], Robert Sansom[2] and Emma Randle[2]

[1] Evolutionary Studies Institute (ESI), School of Geosciences, University of the Witwatersrand, Johannesburg, South Africa
[2] School of Earth and Environmental Sciences, University of Manchester, Manchester, United Kingdom

## ABSTRACT

Ostracoderms (fossil armoured jawless fishes) shed light on early vertebrate evolution by revealing the step-wise acquisition of jawed vertebrate characters, and were important constituents of Middle Palaeozoic vertebrate faunas. A wide variety of head shield shapes are observed within and between the ostracoderm groups, but the timing of these diversifications and the consistency between different measures of their morphospace are unclear. Here, we present the first disparity (explored morphospace) versus diversity (number of taxa) analysis of Pteraspidiformes heterostracans using continuous and discrete characters. Patterns of taxic diversity and morphological disparity are in accordance: they both show a rise to a peak in the Lochkovian followed by a gradual decline in the Middle-Late Devonian. Patterns are largely consistent for disparity measures using sum of ranges or total variance, and when using continuous or discrete characters. Pteraspidiformes heterostracans can be classified as a "bottom-heavy clade", i.e., a group where a high initial disparity decreasing over time is detected. In fact, the group explored morphospace early in its evolutionary history, with much of the subsequent variation in dermal armour occurring as variation in the proportions of already evolved anatomical features. This Early Devonian radiation is also in agreement with the paleobiogeographic distribution of the group, with a maximum of dispersal and explored morphospace during the Lochkovian and Pragian time bins.

Corresponding author
Marco Romano,
marco.romano@uniroma1.it

## INTRODUCTION

Ostracoderms (extinct, bony jawless vertebrates) are a paraphyletic assemblage comprising the jawed vertebrate stem group, which dominated the early vertebrate assemblages, first appearing with high levels of diversity in the Silurian (*Sansom, Randle & Donoghue, 2015*). Seen within the ostracoderms are many novel vertebrate features such as the first appearance of mineralised bone, paired appendages and paired sensory organs (*Donoghue & Keating, 2014*). The diversity of headshield shapes is large, with many groups variously possessing lateral, anterior and dorsal processes. The timing and nature of these morphological diversifications is unclear, as is the best way to quantify the morphological variation. For
example, the difficulty in taxonomic assignment and phylogenetic reconstruction of the Pteraspidiformes (the largest clade of heterostracan ostracoderms) can be attributed to the continuous variation in their dermal plates which is often used to discriminate between taxonomic grades (*Ilyes & Elliott, 1994*; *Pernègre, 2002*; *Pernègre & Goujet, 2007*; *Pernègre & Elliott, 2008*; *Randle & Sansom, 2017a*; *Randle & Sansom, 2017b*). The Pteraspidiformes are characterised by possessing separate dorsal, ventral, rostral and pineal plates along with paired branchial, orbital and in some instances cornual plates (Fig. 1D) (*Blieck, 1984*; *Blieck, Elliott & Gagnier, 1991*; *Janvier, 1996*; *Pernègre & Elliott, 2008*; *Randle & Sansom, 2017a*; *Randle & Sansom, 2017b*). The Pteraspidiformes include many families and taxa of uncertain affinities. The Anchipteraspididae and *Protopteraspis* are stratigraphically the oldest Pteraspidiformes first occurring in the Pridoli (*Elliott, 1983*; *Blieck, 1984*; *Blieck & Tarrant, 2001*). The Anchipteraspididae and *Protopteraspis* are both small Pteraspidiformes with blunt shaped rostra (Fig. 1E). The Anchipteraspididae have a few anatomical differences to the remaining Pteraspidiformes including; a pineal plate enclosed within their dorsal plate, rather than positioned between the rostral and dorsal plates as seen in all other Pteraspidiformes, a fused orbito-cornual plate (with are completely separate in other Pteraspidiformes taxa) and the centre of growth in the dorsal plate anterior to the midline, whereas, in other forms it is centrally or posteriorly positioned (*Randle & Sansom, 2017a*; *Elliott, 1983*). Other families include the Rhinopteraspididae (Fig. 1E), which contains taxa with extremely lengthened rostra and headshields e.g., *Rhinopteraspis* and *Althaspis*, the Protaspididae, which contains taxa with widened headshields and forms with posteriorly extended branchial plates and absent cornual plates, and finally the Doryaspididae, containing the enigmatic *Doryaspis,* which has an unusually dorsally orientated mouth, extreme laterally extended cornual plates and a unique pseudorostum (*White, 1935*; *Janvier, 1996*; *Pernègre, 2002*). *Randle & Sansom (2017a)* also found the two Psammosteidae taxa to be nested within the Pteraspidiformes. The Psammosteidae are stratigraphically the youngest heterostracans and are characterised by having a dorsally orientated mouth and small 'platelets' separating their major plates (*Blieck, 1984*; *Janvier, 1996*) (Fig. 1E).

Due to the Pteraspidiformes possessing a rather uniform anatomy, inclusion of taxonomically informative quantitative data, including the relative sizes and dimensions of dermal plates, was explored in the phylogenetic analyses of *Randle & Sansom (2017a)*, who included two different treatments of quantitative ratio data in their phylogenetic analyses of the Pteraspidiformes. The first treatment discretised the quantitative data into ordinal discrete character states by identifying gaps between the differences of ordered ratio data (>2 standard deviations of the gap data) to infer changes in character states. The second treatment used the raw continuous quantitative data to reconstruct their evolutionary relationships. Inclusion of quantitative data greatly improved the resolution of Pteraspidiformes relationships using traditional discrete characters—however, the two methods provided different and conflicting evolutionary relationships.

One of the goals of this study is to explore morphospace occupation through time using both classic discrete cladistic characters and quantitative continuous characters, along with

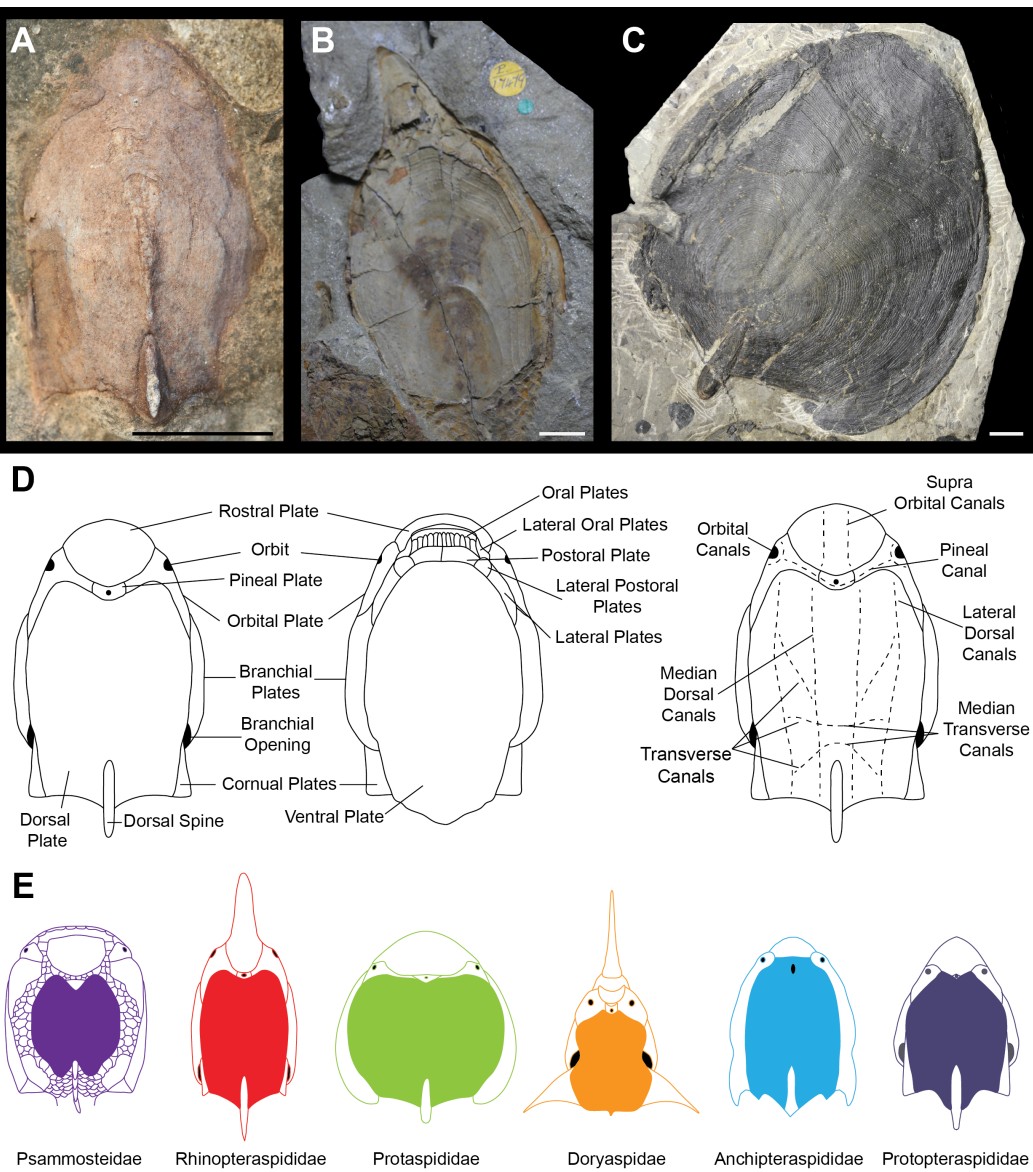

**Figure 1** **Headshield morphologies of pteraspidiform heterostracans (stem-gnathostomes).** (A) Dorsal shield of *Protopteraspis sartoki* NMC.13869 (National Museum of Canada, Ottawa, Canada) a Protopteras-pididae Pteraspidiformes. (B) Ventral view of *Errivaspis waynensis* P.17479 (Natural History Museum, London, UK) a Rhinopteraspididae Pteraspidiformes. (C) Dorsal view of *Cosmaspis transversa* PF4924 (Field Museum, Chicago, USA) a Protaspididae Pteraspidiformes. (D) Pteraspidiformes anatomy. (E) Re-construction cartoons of the main Pteraspidiformes groups and general morphologies. Scale bar −10 mm. Rhinopteraspididae −*Blieck, 1984*, Anchipteraspididae *Elliott, 1984*, Protopteraspididae *Blieck & Tarrant, 2001*.

the signal these phylogenetic morphospace plots provide for the different taxonomic clades within the Pteraspidiformes.

The use of cladistic or more traditional morphometric characters is debated in macro-evolution fields with authors arguing that discrete and morphometric characters differ
in relative degree of independence, homology of the considered features, their rates of evolution and on the nature of the variation being captured (*MacLeod, 2002*; *Klingenberg & Gidaszewski, 2010*). *Mongiardino Koch et al. (2017)* when exploring the scorpion genus *Brachistosternus* using morphospaces derived from discrete and morphometric characters, found the signal derived from these two different data types to be significantly different from each providing a non-congruent picture of their early evolution. For example, their discrete data displayed an 'early burst' scenario, whereas their morphological data did not, which they concluded was due to evolution being driven by species-specific adaptations of morphometric traits. On the contrary, several studies have shown empirically how the results derived from discrete and morphometric characters are fully compatible, providing the same signal on a macro-evolutionary scale (e.g., *Villier & Eble, 2004*; *Anderson & Friedman, 2012*; *Foth, Brusatte & Butler, 2012*; *Hetherington et al., 2015*; *Romano, Brocklehurst & Fröbisch, 2017*). In particular, *Villier & Eble (2004)* were the first to empirically demonstrate that disparity calculated using morphometric measures and discrete characters converge to the same signal, using the echinoid order Spatangoida as a case study. As a general conclusion, the authors stressed how the choice of different morphometric scheme, temporal scale, and taxonomic level does not seem to affect major macroevolutionary trends in disparity.

*Foth, Brusatte & Butler (2012)* demonstrate that different proxies for disparity of pterosaur crania (geometric morphometrics, limb proportions and classic discrete characters) converge on a common macroevolutionary signal. Such results are encouraging in that one metric may be representative of other types of data.

In an exploratory study of disparity focusing on caecilian amphibians, *Hetherington et al. (2015)* found no impact on relative inter-taxon distances when different coding strategies for cladistic characters were considered or by taking in consideration revised concepts of homology. The authors stressed how their results indicate that cladistic and geometric morphometric data seem to carry the same disparity signal, thus summarizing in comparable ways the morphological variation for the clade. The authors in conclusion strongly supported the cladistic dataset as a source from which to calculate and characterize clade disparity.

*Romano, Brocklehurst & Fröbisch (2017)* demonstrated using data from captorhinids that disparity calculated using cladistic discrete characters and continuous morphometric characters, converge to the same macroevolutonary signal through the whole evolutionary history of the group. Interestingly, while the discrete dataset is built essentially on classical cranial characters, the morphometric ones are based almost totally on long bones. As already stressed by *Foth, Brusatte & Butler (2012)*, in the absence of one of the possible proxies, the disparity calculated based on just one type of character can be considered representative of the disparity pattern on a large macroevolutionary scale.

In discussing to what extent the conclusions obtained from their particular study on caecilian amphibians were generalizable, *Hetherington et al. (2015)* strongly encouraged similar studies on other clades, both of invertebrates and vertebrates. In this framework, the specific clade of Pteraspidiformes therefore represents a new interesting case to
empirically test the possible congruence between the signals contained in the discrete and morphometric characters.

Important in this context is whether timing of morphospace occupations as either early or late in the history of a clade and how they compare to changes in taxic diversity. Studies of morphospace occupation in both invertebrates (e.g., *Foote, 1994*; *Foote, 1999*; *Lofgren, Plotnick & Wagner, 2003*; *Villier & Eble, 2004*; *Lefebvre et al., 2006*; *Al-Sabouni, Kucera & Schmidt, 2007*; *Scholz & Hartman, 2007*; *Glaubrecht, Brinkmann & Pöppe, 2009*; *Whiteside & Ward, 2011*; *Deline & Ausich, 2011*; *Bapst et al., 2012*; *Hopkins, 2013*; *Romano et al., 2018*) and vertebrates (e.g., *Prentice, Ruta & Benton, 2011*; *Benson, Evans & Druckenmiller, 2012*; *Ruta et al., 2013*; *Colombo et al., 2015*; *Marx & Fordyce, 2015*; *Larson, Brown & Evans, 2016*; *Romano, 2017a*; *Romano, Brocklehurst & Fröbisch, 2017*) have reconstructed the timing of radiations, with many identifying maximum disparity at the beginning of their evolutionary history (termed 'bottom heavy'), followed by stabilization and constant decrease until their subsequent extinction (e.g., *Gould, Gilinsky & German, 1987*; *Foote, 1992*; *Foote, 1994*; *Foote, 1995*; *Foote, 1999*; *McGhee Jr, 1995*; *Wagner, 1995*; *Smith & Bunje, 1999*; *Eble, 2000*; *Huntley, Xiao & Kowalewski, 2006*; *Ruta et al., 2013*; *Marx & Fordyce, 2015*; *Romano, 2017a*). Here we test the timing of morphospace radiations for Pteraspiformes and compare that to taxic diversity. We compare total variance or as a sum of ranges as measures of disparity, in both discrete and continuous sub-datasets.

## MATERIAL AND METHODS

### Taxa

The analysis was conducted using the phylogenetic analysis dataset of Pteraspidiformes heterostracans recently published by *Randle & Sansom (2017a)*. For the study only the 49 in-group taxa of the original dataset were considered as follows: *Alaeckaspis*, *Althaspis*, *Anchipteraspis*, *Blieckaspis*, *Brachipteraspis*, *Canadapteraspis*, *Cosmaspis*, *Cyrtaspidichthys*, *Djurinaspis*, *Dnestraspis*, *Doryaspis*, *Drepanaspis*, *Errivaspis*, *Escharaspis*, *Eucyclaspis*, *Europrotaspis*, *Gigantaspis*, *Helaspis*, *Lamiaspis*, *Lampraspis*, *Larnovaspis*, *Loricopteraspis*, *Miltaspis*, *Mylopteraspis*, *Mylopteraspidella*, *Oreaspis*, *Palanasaspis*, *Panamintaspis*, *Parapteraspis*, *Pavloaspis*, *Pirumaspis*, *Podolaspis*, *Protaspis*, *Protopteraspis gosseleti*, *Protopteraspis primaeva*, *Psammosteus*, *Psephaspis*, *Pteraspis*, *Rachiaspis*, *Rhinopteraspis*, *Semipodolaspis*, *Stegobranchiaspis*, *Tuberculaspis*, *Ulutitaspis*, *Unarkaspis*, *Woodfjordaspis*, *Xylaspis*, *Zascinaspis carmani*, *Zascinaspis heintzi*. The taxa *Anglaspis*, *Athenaegis* and *Nahanniaspis* chosen as outgroups by *Randle & Sansom (2017a)* were not considered for the study of diversity and disparities through time. Thus, apart from the four species *Protopteraspis gosseleti*, *Protopteraspis primaeva*, *Zascinaspis carmani* and *Zascinaspis heintzi*, the majority of taxa are considered at the genus level. *Foote (1995)* and *Foote (1996)* has empirically shown how analysis conducted at the species and genus level provide equivalent signal (however Smith & Lieberman, 1999 consider the species level as preferable).

### Diversity and disparity

To perform the analysis the following six time bins were selected spanning from the Upper Silurian to the Upper Devonian: Pridoli, Lochkovian, Pragian, Emsian, Eifelian,

Givetian-Frasnian. The Givetian and Frasnian stages were considered in a single time bin, since for the analysis of the disparity at least two taxa must be present in each considered interval. The distribution of taxa in the different time bins was based on the time calibrated tree of Pteraspidiformes heterostracans provided by *Randle & Sansom* (*2017a*, p. 595, fig. 7); the occurrence of taxa for each time bins is reported in supplementary material (Appendix S1).

Taxic diversity for Pteraspidiformes heterostracans is simply the sum of taxa in each time bin. Two disparity analyses were conducted; one on the classical discrete characters and the second, using the continuous characters only. Disparity was calculated both as the total variance and as the sum of ranges for the two different datasets (discrete and continuous). According to several authors (*Foote, 1997*; *Erwin, 2007*; *Ruta, 2009*; *Prentice, Ruta & Benton, 2011*) disparity as total variance indicates essentially how the considered taxa are dispersed in the morphospace, whereas disparity as sum of ranges represents a good indication of the total occupied morphospace through time (see *Wills, Briggs & Fortey, 1994*; *Prentice, Ruta & Benton, 2011*). These indications must be carefully taken into account in the interpretation of the results obtained with the study (see below).

Disparity analysis of the discrete dataset (65 discrete characters, see Appendix S2) (*Randle & Sansom, 2017a*) was subjected to a Principal Coordinates Analysis on the free software PAST 3.10 (*Hammer, Harper & Ryan, 2001*), using the 'Gower' similarity index ($c = 2$ Transformation Exponent), preferable to the simple Euclidean distance (see *Hammer, 2013*). Coding for the discreet character 42 in *Helaspis* and *Psephaspis* has been replaced by a question mark being polymorphic in the two taxa (two states of the character present). The PCO scores were used to calculate disparity, both as total variance and as sum of ranges, for the discrete character dataset (see Appendix S1). Only the first 23 principal coordinates were considered in the results, as the 24th was constant, not contributing to disparity.

22 continuous characters from *Randle & Sansom* (*2017a*; see Appendix 2) were analysed using a Principal Component Analysis, again using the software PAST 3.10. Missing entries were computed using the 'iterative imputation' in PAST, as suggested by *Hammer (2013)*. Before the analysis, the raw data were log transformed for the correspondence of the log-transform to an isometric null hypothesis and to fit linear models (see *Chinnery, 2004*; *Cheng et al., 2009*; *Romano & Citton, 2015*; *Romano & Citton, 2017*; *Romano, 2017b*; *Citton et al., 2017*). Linear measures are in general preferable to ratios in Principal Component Analyses (see *Hammer & Harper, 2006*). However in this case the original ratios were used to perform the analysis, to be congruent with the results obtained by *Randle & Sansom (2017a)*. Even in this case, the scores obtained from the 22 principal components were used to calculate disparity both as sum of ranges and variance (see Appendix S1).

## RESULTS

The first occurrence of Pteraspidiformes heterostracans is in the Pridoli (Upper Silurian) with fairly low levels of diversity (Fig. 2A), and the clade is represented by just 4 genera. However, their diversity rises and attains its maximum in the Lochkovian to Pragian. From

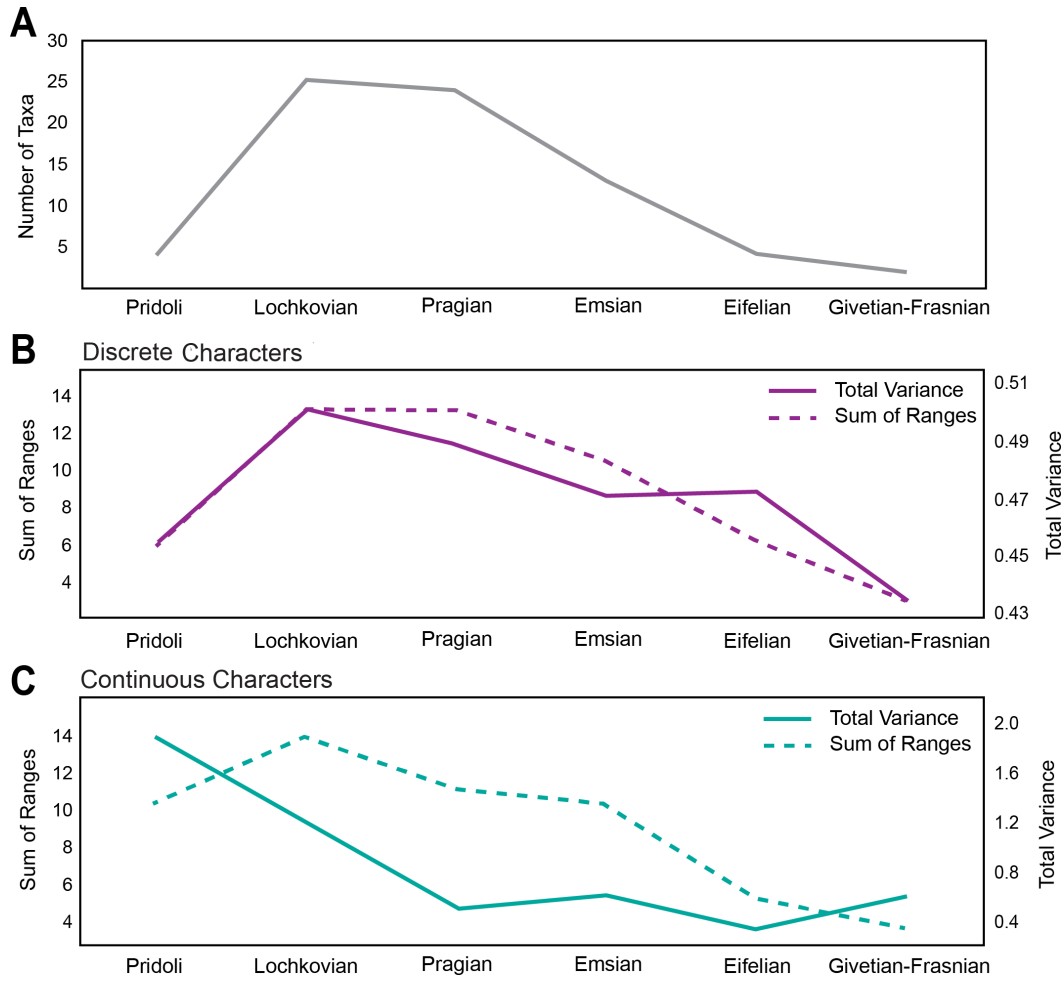

**Figure 2** **Disparity and diversity trends in stem gnathostomes pteraspidiformes heterostracans.** (A) Pteraspidiformes taxic diversity through time, (B) disparity of Pteraspidiformes heterostracans using discrete characters (both total variance and sum of ranges), (C) disparity of Pteraspidiformes heterostracans using continuous characters (both total variance and sum of ranges).

the Emsian onwards the number of taxa begins to decrease consistently until their demise in the Frasnian (Upper Devonian).

Disparity for the discrete characters (Fig. 2B) follow a very similar pattern to diversity, especially the sum of ranges. Disparity, as measured by total variance, begins to decrease in the Pragian, while the sum of ranges disparity remains at the same level of the preceding time bin (i.e., Lochkovian). Sum of ranges disparity begins to decline from the Emsian onwards mirroring that of diversity; however, disparity as total variance shows the same value for the Emsian and Eifelian after which it decreases abruptly until it reaches the minimum in the Givetian-Frasnian.

Similarly to the discrete characters the trend of disparity as sum of ranges for the continuous characters (Fig. 2C), closely matches the diversity through time except for a peak in the Pragian. Contrasting with the discrete characters, the continuous characters

have high levels of disparity (for both sum of ranges and total variance) in the Pridoli. Disparity as total variance is decoupled with respect to diversity, with maximum disparity occurring at the beginning of their evolutionary history rather than in the Lochkovian, as seen in the discrete characters disparity. After this initial peak in the Pridoli, disparity declines until the Pragian and remains low until Givetian-Frasnian.

Morphospace occupation for the discrete characters and continuous characters through time can be seen in Fig. 3A. Maximum morphospace exploration (convex hull area) for the discrete characters is observed in the Lochkovian, which overlaps with morphospace occupied by Pteraspidiformes in the Pridoli and subsequent time bins (Pragian-Frasnian). Figure 3B shows morphospace occupation of Pteraspidiformes as described by the continuous characters. There appears much more overlap in morphospace occupation through time bins than seen in the discrete characters, with one taxon extending morphospace occupation in the Pridoli. Throughout the majority of their history the Pteraspidiformes, occupy similar morphospace.

Figure 4 shows the relative position of Pteraspidiformes taxa, grouped by family, in their Principal component analyses (continuous characters) and Principal coordinates (discrete characters) using the first two axes. There is much overlap in Pteraspidiformes morphospace using the continuous characters (Fig. 4A), whereas, the discrete morphospace plot (Fig. 4B) shows less overlap between the taxonomic groups. In particular, the Doryaspidae and Anchipteraspididae are very well separated, without overlap from the convex hulls of other families in the continuous character plot.

Other patterns seen in the continuous character plot (Fig. 4A) includes the Protopteraspididae overalpping with all the other convex hulls, with a truly substantial superimposition with the Anchipteraspididae, which in this case are not well separated from morphospaces explored by other groups. Another interesting result is that members of Psammosteidae do not cluster together in the graph, with *Psammosteus* occurring completely within the morphospace of the Doryaspidae. Many Pteraspidoidei *incertae sedis* fall within the convex hull identified by the families recognized by *Randle & Sansom (2017a)*; the only taxa that fall outside a convex hull or the overlapping of several convex hulls are *Eucyclaspis*, *Parapteraspis*, and *Podolaspis*.

The scatter plot of the PCA conducted on discrete characters is shown in Fig. 4B. patterns include overlap between the Rhinopteraspididae and Protopteraspididae, with *Althaspis* occurring in the shared morphospace. A second overlap in morphospace occupation is observed in the ranges of Protaspididae and Protopteraspididae, with *Tuberculaspis* and *Lampraspis* falling well inside the morphospace of Protopteraspididae. Among the Pteraspidoidei *incertae sedis*, the taxa *Djurinaspis*, *Dnestraspis*, *Europrotaspis*, *Lamiaspis*, *Larnovaspis*, *Oreaspis*, *Pteraspis Semipodolaspis* and *Unarkaspis* are not included in any convex hull identified by the PCA; *Mylopteraspis*, *Eucyclaspis* fall within the Protaspididae; *Alaeckaspis*, *Blieckaspis*, *Eucyclaspis*, *Mylopteraspidella*, and *Protaspis* fall within the morphospace of Protopteraspididae; *Parapteraspis* and *Pirumaspis* fall within the convex hull identified by the Rhinopteraspididae. Compared to the result obtained with the continuous characters (Principal Component Analysis), a greater and substantial separation is evident among the families of Pteraspidiformes in morphospace.

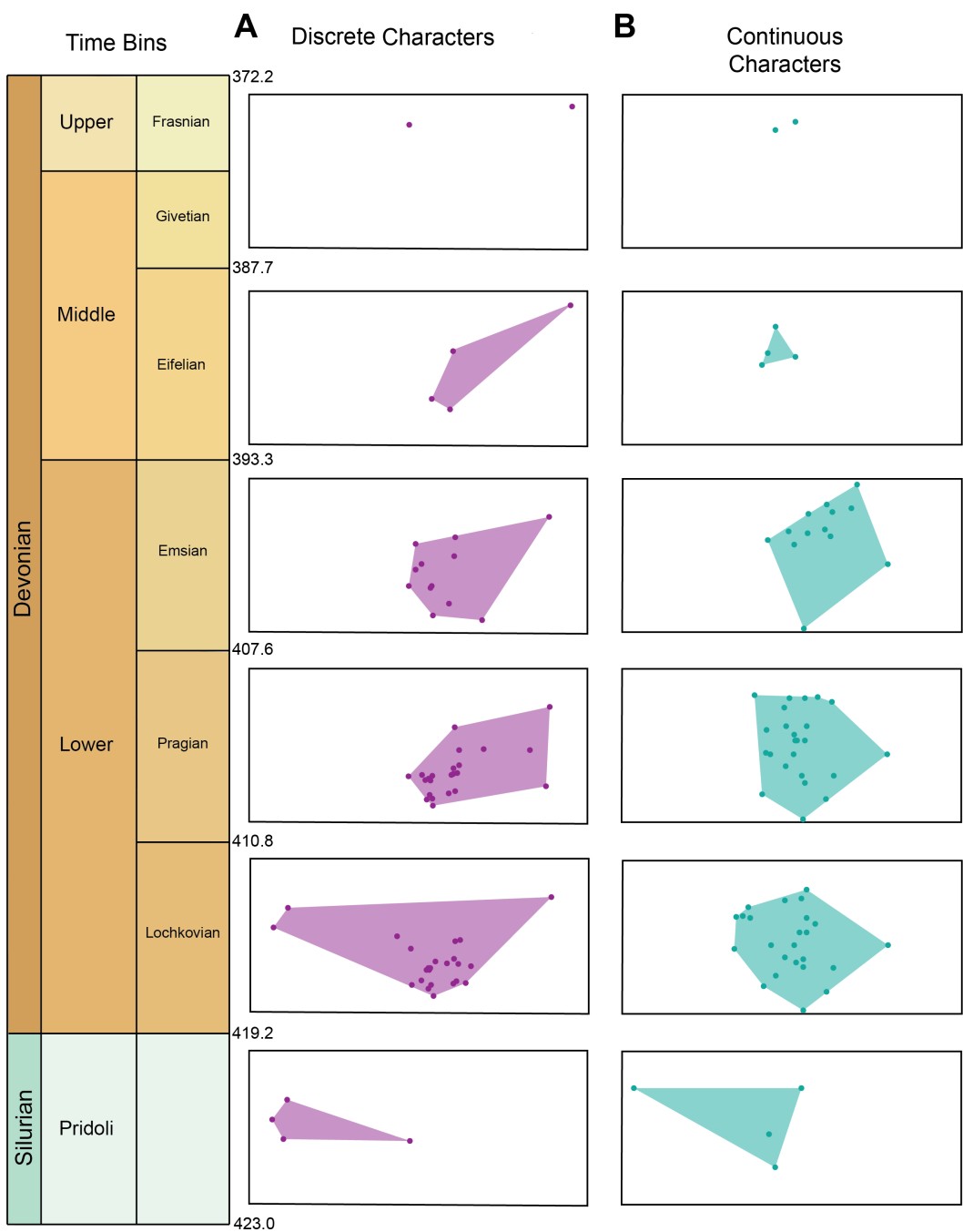

**Figure 3** **Morphospace occupation for the discrete characters and continuous characters through time in pteraspidiformes heterostracans.** Morphospace occupation through time in Pteraspidiformes heterostracans for the (A) discrete characters, and (B) continuous characters.

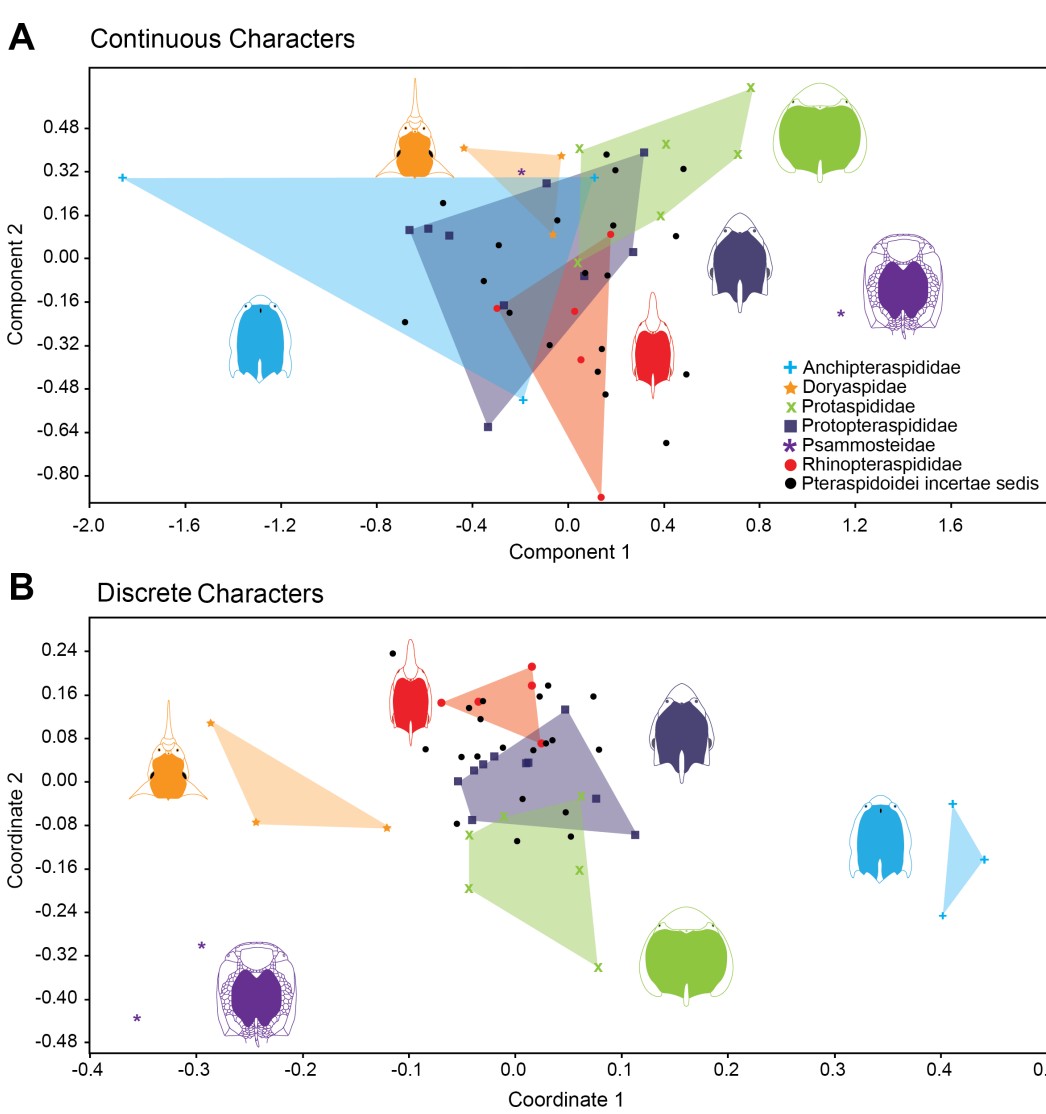

**Figure 4** **Morphospace exploration for continous and discrete characters in pteraspidiformes heterostracans.** Scatter plots of first two principal components performed on continuous characters (A) and first two principal coordinated on discrete characters (B). The groups are named following *Randle & Sansom (2017a)*.

## DISCUSSION

Diversity-disparity curves show that the disparity as sum of the ranges and total variance provide completely compatible and mostly superimposable macroevolution trends for classical discrete characters (Fig. 2), whereas for the continuous ones the trend detected by the total variance results is quite different. Considering total variance as a measure of the dispersion of taxa (*Foote, 1997*; *Erwin, 2007*; *Ruta, 2009*; *Prentice, Ruta & Benton, 2011*), the result shows that for continuous characters, the maximum dispersion in the morphospace is found at the beginning of Pteraspidiformes evolutionary history (during the Pridoli); the dispersion then decreased consistently from the Lochkovian

onwards. A possible explanation for this trend in the total variance could be the 'early burst' scenario. *Mongiardino Koch et al. (2017)* suggest that an 'early burst' result can be spurious if obtained from cladistic discrete characters as these are biased towards obviating autapomorphic characters overestimating evolution at the base of a clade. We, however, identify this pattern in our continuous dataset, perhaps suggesting that any generalization must be taken with caution, and that different clades can react differently and peculiarly to disparity analysis.

For all the above, worthy of note is a brief discussion on the reliability of using classical discrete cladistic characters to investigate disparity trend in a clade. *Anderson & Friedman (2012)*, on the basis of an empirical study on early gnathostomes, highlighted possible inconsistencies between the signals obtained from discrete and morphometric characters. In particular, according to the authors, the biggest issue with cladistics characters for disparity analyses derives from the exclusion of autapomorphies from the original matrix (as not informative for phylogeny), and of potentially undersampling 'noisy' homoplastic features. These elements could obviously lead to the loss of information to reconstruct the total morphospace of a group during its evolutionary history. However, the inconsistency of the results obtained on early gnathostomes by *Anderson & Friedman (2012)* is strictly related to specific functional variation in the clade, and not to the overall morphological disparity. In fact the authors consider in general the disparity based on cladistics characters as "*an important and broadly applicable tool for quantitative paleobiological analyses*" (*Anderson & Friedman, 2012*, p. 1262), even if not really suitable for ecological and functional variation analyses. The same authors stressed how disparity analyses conducted on a cladistic dataset will in any case be characterized by a cladistic signal that needs to be acknowledged when they are used. Our analysis of disparity using discrete cladistic data does indeed characterise cladistic signal (Fig. 4A), but the results are also in agreement with analysis of disparity using non-cladistic continuous data, for sums of ranges at least (Fig. 2). In any case, in the interpretation of the results in the present paper, and in numerous other contributions based on cladistic characters, we must bear in mind that several homoplastic characters, autapomorphies and background 'morphological noise' will be missing from cladistic datasets, so most probably underestimating the 'total disparity' for a clade (however in the dataset used in the present contribution some autapomorphies are considered, i.e., character 4 in *Djurinaspis*, characters 22 and 58 in *Doryaspis*, character 52 in *Miltaspis*, character 60 in *Lamiaspis*). It follows that part of the original biological variation, expressed as disparity, will be missing from the cladistics dataset analyses. However, *Hetherington et al. (2015)* even after obtaining the same large scale trend in disparity from discrete and morphometric characters, strongly preferred discrete cladistic character data since "*in addition to encompassing the gain and loss of structures, they readily allow all aspects of organismal biology to be captured, as opposed to morphometrics which, for entirely practical reasons, is invariably only ever applied to proxy components of anatomy*" (*Hetherington et al.* (*2015*, p. 398).

The results in general indicate that Pteraspidiformes heterostracans explored morphospace early in their evolutionary history (Pridoli-Lochkovian), with much of the subsequent variation in their dermal armour occurring as variation in the proportions

of already evolved anatomical features (Figs. 2 and 3). Considering the total variance as a measure of the dispersion of the taxa in morphospace (see *Foote, 1997*; *Erwin, 2007*; *Ruta, 2009*; *Prentice, Ruta & Benton, 2011*) and the sum of ranges as an indication of the total occupied morphospace (see *Wills, Briggs & Fortey, 1994*; *Prentice, Ruta & Benton, 2011*), the results also indicate that the Pteraspidiformes increase in taxonomic diversity also corresponds to an increase in taxa dispersion in morphospace and morphologies. This is followed by a progressive decrease in taxic diversity and morphospace occupation from the Emsian until their demise in the Frasnian (Fig. 2). Indeed the disparity trajectories closely follow taxonomic diversity throughout their history.

Extending the classic diversity categories identified by *Gould, Gilinsky & German (1987)* to morphospace exploration, the Pteraspidiformes constitute a "bottom-heavy clade", i.e., a group where a high initial disparity decreasing over time is detected. The great initial disparity in this case does not coincide with the evolutive first appearance of the group but it is shifted by at least one stage forward. An early radiation with a maximal disparity at the beginning of the evolutionary story of a clade had been found empirically in the literature for example for blastozoans (*Foote, 1992*), brachiopods (*Carlson, 1992*; *McGhee Jr, 1995*; *Smith & Bunje, 1999*), Neoproterozoic acritarchs (*Huntley, Xiao & Kowalewski, 2006*), Palaeozoic gastropods (*Wagner, 1995*), and crinoids (*Foote, 1994*; *Foote, 1995*; *Foote, 1999*). In the same way, a decrease in occupied morphospace during the evolutionary history of a clade was found for example in Carboniferous ammonoids (*Saunders & Work, 1996*; *Saunders & Work, 1997*), rostroconchs (*Wagner, 1997*) and Palaeozoic stenolaemate bryozoans (*Anstey & Pachut, 1995*).

The discrete and continuous characters display differing patterns of overall morphospace occupation for the different taxonomic groups (Fig. 4). The continuous characters displays much overlap of taxonomic groups in morphospace, whereas, the discrete dataset show separate morphospace occupation for the families recognized by *Randle & Sansom (2017a)*, apart from a slight overlap in the convex hulls of Protaspididae and Protopteraspididae, and between Protopteraspididae and Rhinopteraspididae.

## CONCLUSIONS

In this paper we present the first disparity (explored morphospace) versus diversity (number of taxa) analysis of Pteraspidiformes heterostracans using continuous and discrete characters. Patterns of morphological disparity and taxic diversity are in accordance, both showing a rise to a peak in the Lochkovian followed by a gradual decline in the Middle-Late Devonian.

The Pteraspidiformes, unlike other groups of heterostracans (i.e., Cyathaspididae and Traquairaspididae) arose later in the evolutionary history of the Heterostraci (the first heterostracans are from the Wenlock) (*Randle & Sansom, 2017a*; *Dineley & Loeffler, 1976*). Therefore, it is less likely that the early history of the Pteraspidiformes clade is lost due to fossil record or other abiotic biases, such as sea-level, as seen with other ostracoderm clades (*Sansom, Randle & Donoghue, 2015*). There is good correspondence between maximum taxonomic diversity and saturation of occupied morphospace, identifying the

Pteraspidiformes heterostracans as a 'bottom' heavy clade, with most structural 'bauplans' and major morphologies already explored by the group in the Early Devonian.

## ACKNOWLEDGEMENTS

We would like thank museum staff for access to specimens, the Editor and two anonymous reviewers who greatly improved the manuscript.

### Funding

Part of this work was made possible by financial support (Marco Romano) from the DST/NRF Centre of Excellence for Palaeosciences (CoE in Palaeosciences). There was no additional external funding received for this study. The funders had no role in study design, data collection and analysis, decision to publish, or preparation of the manuscript.

### Grant Disclosures

The following grant information was disclosed by the authors:
Centre of Excellence for Palaeosciences.

### Competing Interests

The authors declare there are no competing interests.

### Author Contributions

- Marco Romano conceived and designed the experiments, performed the experiments, analyzed the data, contributed reagents/materials/analysis tools, prepared figures and/or tables, authored or reviewed drafts of the paper, approved the final draft.
- Robert Sansom analyzed the data, authored or reviewed drafts of the paper, approved the final draft.
- Emma Randle analyzed the data, prepared figures and/or tables, authored or reviewed drafts of the paper, approved the final draft.

### Data Availability

The dataset for Pteraspidiformes analysis is provided as a Supplemental File.

### Supplemental Information

Supplemental information for this article can be found online at http://dx.doi.org/10.7717/peerj.5249#supplemental-information.

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
