# Peer review of "Morphospace saturation in the stem-gnathostomes pteraspidiformes heterostracans: an early radiation of a ‘bottom’ heavy clade"

_PeerJ, doi:10.7717/peerj.5249_

## Round 0.1 · original submission · Major Revisions

Both reviewers identified some major issues with this study that need resolving before it is ready for publication, in particular, the problem of use discrete characters originally selected for phylogenetic analysis in disparity analyses, which weakens the validity of your conclusions. This issue was investigated by Anderson & Friedman (2012; JVP 32: 1254-1270) and should be considered here.

The other issue is one of detail. There is little, if any, consideration of morphology in this study of morphospace occupation. The discussion would be greatly improved by an explanation of the morphological meaning of your distributions, and overall it would be very helpful to have pictures of some of the taxa under study. I understand that you are referring us back to a previous publication for the character list, but it would be good to have at least some description of characters, otherwise the whole study becomes rather abstract.

I think the issues identified by the reviewers will require a little thought, so I have given the decision of major revisions. I will look forward to seeing a revised version of the manuscript.

Reviewer 1 ·

Basic reporting

no comment

Experimental design

The authors utilize data from a previous systematics paper for the disparity analyses. The discrete data included here is presumably cladistics data used to build the phylogeny of the previous paper. The use of cladistics characters undermines one of the stated goals of the paper: to test whether the use of discrete or continuous characters show similar results for morphological disparity. The problem is that the discrete characters used here were originally selected for their value to phylogenetic analyses. Not surprisingly, when the morphospaces are plotted, the discrete characters separate the various taxonomic groups into distinct regions, the continuous characters do not. The authors discuss why this might be and suggest more convergence amongst continuous characters than discrete ones. However, the more likely answer is that the discrete characters were originally selected to separate clades not quantify overall variation. Previous published work has discussed this aspect of cladistics characters as phylogenetic analyses actively remove characters that would be considered convergent, and so fail to fully describe morphology. This needs to be discussed in the paper as I don’t not believe that the difference between discrete and continuous characters is about the form of the data, but about what the data was originally collected for.

Validity of the findings

I disagree that the data supports some of the conclusions being drawn by the authors. Specifically, the conclusion that there is concordance between the taxonomic and morphological data. These data are compared in four permutations illustrated in figures 1 and 2. The first two compare the discrete morphological data to diversity, which do show concordance. However, given the strong phylogenetic signal within the cladistics discrete data used, this is not surprising. For the continuous data, only the sum of ranges showed concordance. The total variance dataset shows almost opposite trends between the two. This is mentioned in passing in the discussion, but should be given more attention. At the very least, the data is not fully supportive of the concordance hypothesis.

Additional comments

1. Pg. 6; Ln 119: How many continuous characters were used?
2. Pg. 7; Ln 128: In line with the previous comment, I am unclear how you get 22 PCs from your analysis. Were there 22 original variables?
3. Pg. 8; Ln 155: The text mentions that the Pridolian forms occupied a different area of space than the rest of the taxa in later time bins for continuous characters. However, looking at figure 3, there is quite a bit of overlap in the center of the space between these time periods. There is certainly a shift in space, but there is common shapes being seen.
4. Figure 4: Just a stylistic suggestion, but it would be good to have some silhouettes showing the shapes of these groups, to give context to the morphospace.

Reviewer 2 ·

Basic reporting

There is not much background provided on the clade, nor are there any images of headshields or visualization of characters in the main text. Thus, this study is difficult to follow for readers not already intimately familiar with the previous study which generated this dataset (Randle and Sansom, 2017) and the various genera and families within Heterostraci. There should be at least some images of fishes, perhaps from different parts of morphospace.

In addition, there is no reporting of what the main components plotted in Figs. 3 and 4 represent, making it difficult to interpret what the relative positions of different groups might mean. The authors make numerous points about separation and overlap between different clusters (131-183) without explaining what this means morphologically or why it might be unexpected.

Experimental design

It is not clear why discrete characters were used to compare disparity and diversity through time. Diversity in this case is the number of terminal taxa in each bin, and the discrete characters were chosen to distinguish these taxa in a phylogenetic analysis. Thus we should expect the discrete character combinations to approximate the number of terminals in each bin. We should also expect clustering in morphospace that mirrors the tree structures, since clades will share character states (synapomorphies) that distinguish them from other families. Both these results are reported here.

Thus the discrete results come from structural biases of cladistic data, which is constructed to test phylogenetic rather than gross morphological, evolutionary or ecological hypotheses. Thus, I am not sure the use of discrete cladistic characters is appropriate or directly relevant to the aims of this study, or if the results of these analyses have any bearing on the hypotheses here, particularly given conflicting results from continuous characters.

In fact, conflicts between taxon placement in morphospaces produced by cladistic and traditional morphometric characters for scorpions have been used to argue against the use of cladistic datasets for testing hypotheses on disparity, as phylogenetic signal rather than meaningful morphological variation drives the results (Koch et al. 2017, Journal of Evolutionary Biology)

Validity of the findings

I do not think the discrete character results are useful or valid for the reasons given above.

The continuous character dataset suggests convergence between different groups, and higher disparity early on, which is the main finding of the paper. These results should be presented up front.

The discussion of biogeography and other aspects of heterostracan evolution in the conclusion seem perfunctory, and are not directly related to the results.

---

## Round 0.2 · Major Revisions

Please find attached the reviewers’ comments on your revised manuscript. You will see that there is still some disagreement between the reviewers on the validity of the methods and the findings. Reviewer 1 is now largely satisfied that you have addressed all their comments, although they would like to you to make a few further changes with regard to the interpretation of your results. However, reviewer 2 is less satisfied – they still have considerable concerns about the use of cladistic characters to produce morphospaces and the value of comparing these to taxonomic diversity.

I have some sympathies with reviewer 2’s comments myself. I’m not so concerned with the value of the analysis – PeerJ articles are only judged on methodological and scientific soundness, not impact or novelty – but I think the discussion needs to be very clear what interpretations can and can’t be drawn from your results. The most important thing to add is a clear statement that, even when using the continuous characters, there will still be phylogenetic signal in the results, and an acknowledgement that by using cladistic characters (whether discrete or continuous) you are still excluding autapomorphic characters that contribute to overall morphological disparity. By emphasising these points you will necessarily need to be much more cautious in your interpretation of results – your results potentially point to an early burst scenario, but you can’t be certain about this because of the data you analysed. I think there should also be some indication of why you didn’t use more standard morphometric techniques i.e. landmark-based methodologies.

Potentially, the more interesting outcome from your study is not so much what it tells us about pterapsidiforms, but what it can tell us about the use of discrete vs cladistic characters. To me, the first line of your discussion is misleading when it says that sum of ranges and total variance provide “completely compatible and mostly superimposable macroevolution trends for classical discrete and continuous characters” – the trend for total variance of continuous characters looks very different to me. You do go on to note this, but I would modify this first sentence I think.

The other thing that needs adding is a list of your discrete and continuous characters. These could be added to the appendix file. I think that as they are so crucial to your analysis, it is not sufficient to simply direct the reader to another paper to find out what they are.

I don’t understand the phrase ‘discrete-with-discretised characters’. Surely ‘discretised characters’ will suffice?

Lastly, I have made some typographical edits to the attached PDF.

I hope all of the above is clear. I look forward to seeing a revised version of the manuscript in light of the reviewers’ and my comments.

Reviewer 1 ·

Basic reporting

No Comment

Experimental design

I appreciate the authors taking the time to investigate previous literature on the nature of cladistic vs. continuous characters in order to better frame their work. However, I still have a few minor issues with the discussion, though I believe these are easy to address.

In the discussion, the authors devote a paragraph to summarizing Anderson & Friedman 2012 and conclude that it doesn't pertain to the current study because they weren't trying to ascertain functional signals. It is true that the Anderson & Friedman paper was focused on interpreting function from cladistics characters; however, the authors point out key issues with cladistic characters identified in that paper, which are not restricted to functional analyses. The lack of autapomorphies and removal of noisy, homoplastic characters in cladistics datasets will still be a concern for "total disparity" analyses. These phylogenetically problematic character types are part of the overall variation, but will be missing from cladistic datasets. In other words, variation will still be missing from the cladistics dataset.

The authors also quote Anderson & Friedman saying that cladistic characters are broadly useful. Unfortunately, the authors here have ignored the discussion that followed that quote, which points out that cladistic characters will always have a cladistic signal that needs to be acknowledged when they are used.

None of this undermines this manuscript. However, care needs to be taken to acknowledge that the cladistic data used will have a phylogenetic bias (due to lacking certain types of characters that contribute to overall variation), and the discussion of the results needs to include that.

Validity of the findings

My concerns have been addressed here.

Additional comments

Overall, the authors have addressed my concerns about the manuscript (I especially like the head-plate inclusions in the figures). My only remaining concern is the way they have interpreted some of the previous work on cladistic characters that was suggested. These are minor textual corrections that shouldn't be a problem to fix, but should be addressed.

Reviewer 2 ·

Basic reporting

The authors defends their use of cladistic characters by noting they are only attempting to capture morphological variation without reference to function. They also give more complete explanations of taxa and assumptions.

Experimental design

The methods are explained. That said, I still do not understand the point of comparing diversity of taxonomically relevant character state combinations (as cladistic characters are) to taxonomic diversity, or what this is supposed to show. If the aim is not to capture functional or ecological diversity, but just morphological variation, and this tracks taxonomic diversity as in previous efforts, it is not clear what this study is supposed to demonstrate or why it was done.

Based on one previous study from 2012, the authors suggest that in some contexts, morphological disparity from cladistic characters produces the same signal as morphometric data. Yet they do not test or explain when this occurs, when later studies have produced conflicting results, and what leads them to believe heterostracans are one of those clades in which cladistic characters fully capture variation (even just for the purposes of showing morphological diversity). The head shields of these species are certainly very diverse beyond the traits used for phylogenetic analysis, including various autopomorphic extensions. Do these not contribute to diversity?

Given that, the authors still do not explain why they did not attempt to use morphometrics to more convincingly determine whether they produced the same signal as discrete cladistic characters (given the paucity of previous tests) and to more directly address their question of whether morphological disparity in total matches taxonomic diversity. This study still seems like a minimum effort with minimum relevance.

Validity of the findings

Morphological disparity tracks taxonomic diversity, which is again expected given that taxonomic/phylogenetic differences are driving the signal here (a point which is not fully addressed).

I do not see how the authors infer a surprising 'early burst' pattern given this result - early in this case (pre-mid-Devonian) represents the core of heterostracan evolution history and the peak of their diversity. Likewise, low disparity early in the history of the clade is not unexpected given that there are only 4 taxa. It is entirely possible these four are in fact morphologically disparate in terms of function, ecology, and shape, but all these factors are excluded. likewise, late survivors could be equally disparate to the more numerous heterostracans of the early Devonian along those unsampled axes.

In sum, the limited scope of the analyses, the validity and relevance of which is untested, undermines the ability of the authors to make any conclusions that are not already apparent from raw diversity (i.e. diversity/disparity is low early one, rises in the Silurian, and falls in the mid-Devonian before extinction). There is little value added by the use of morphological data.

---

## Round 0.3 · accepted · Accept

Thank you for dealing with the second round of reviewer comments. I appreciate that it has been difficult to reconcile the opposing points of view, but I think you have managed it, and have made clear the arguments for and against the use of cladistic characters in disparity analyses. I'm now happy to accept the manuscript for publication.

I'm not sure if you can see the attached PDF, but if you can, you'll see that I've made a few typographical corrections. I've asked the production team to sort these out during typesetting, so you shouldn't need to do anything further.

#